# Geometry-Aware Neural Rendering

**Josh Tobin**
OpenAI & UC Berkeley
josh@openai.com

**OpenAI Robotics**[*]
OpenAI

**Pieter Abbeel**
Covariant.AI & UC Berkeley
pabbeel@cs.berkeley.edu

## Abstract

Understanding the 3-dimensional structure of the world is a core challenge in computer vision and robotics. Neural rendering approaches learn an implicit 3D model by predicting what a camera would see from an arbitrary viewpoint. We extend existing neural rendering to more complex, higher dimensional scenes than previously possible. We propose Epipolar Cross Attention (ECA), an attention mechanism that leverages the geometry of the scene to perform efficient non-local operations, requiring only $O(n)$ comparisons per spatial dimension instead of $O(n^2)$. We introduce three new simulated datasets inspired by real-world robotics and demonstrate that ECA significantly improves the quantitative and qualitative performance of Generative Query Networks (GQN) [7].

## 1 Introduction

The ability to understand 3-dimensional structure has long been a fundamental topic of research in computer vision [10, 22, 26, 34]. Advances in 3D understanding, driven by geometric methods [14] and deep neural networks [7, 31, 40, 43, 44] have improved technologies like 3D reconstruction, augmented reality, and computer graphics. 3D understanding is also important in robotics. To interact with their environments, robots must reason about the spatial structure of the world around them.

Agents can learn 3D structure implicitly (e.g., using end-to-end reinforcement learning [24, 25]), but these techniques can be data-inefficient and the representations often have limited reuse. An explicit 3D representation can be created using keypoints and geometry [14] or neural networks [44, 43, 30], but these can lead to inflexible, high-dimensional representations. Some systems forego full scene representations by choosing a lower-dimensional state representation. However, not all scenes admit a compact state representation and learning state estimators often requires expensive labeling.

Previous work demonstrated that Generative Query Networks (GQN) [7] can perform neural rendering for scenes with simple geometric objects. However, robotic manipulation applications require precise representations of high degree-of-freedom (DoF) systems with complex objects. The goal of this paper is to explore the use of neural rendering in such environments.

To this end, we introduce an attention mechanism that leverages the geometric relationship between camera viewpoints called Epipolar Cross-Attention (ECA). When rendering an image, ECA computes a response at a given spatial position as a weighted sum at all *relevant positions* of feature maps from the context images. Relevant features are those on the epipolar line in the context viewpoint.

---

[*]Ilge Akkaya, Marcin Andrychowicz, Maciek Chociej, Mateusz Litwin, Alex Paino, Arthur Petron, Matthias Plappert, Raphael Ribas, Jonas Schneider, Jerry Tworek, Nik Tezak, Peter Welinder, Lilian Weng, Qiming Yuan, Wojciech Zaremba, Lei Zhang

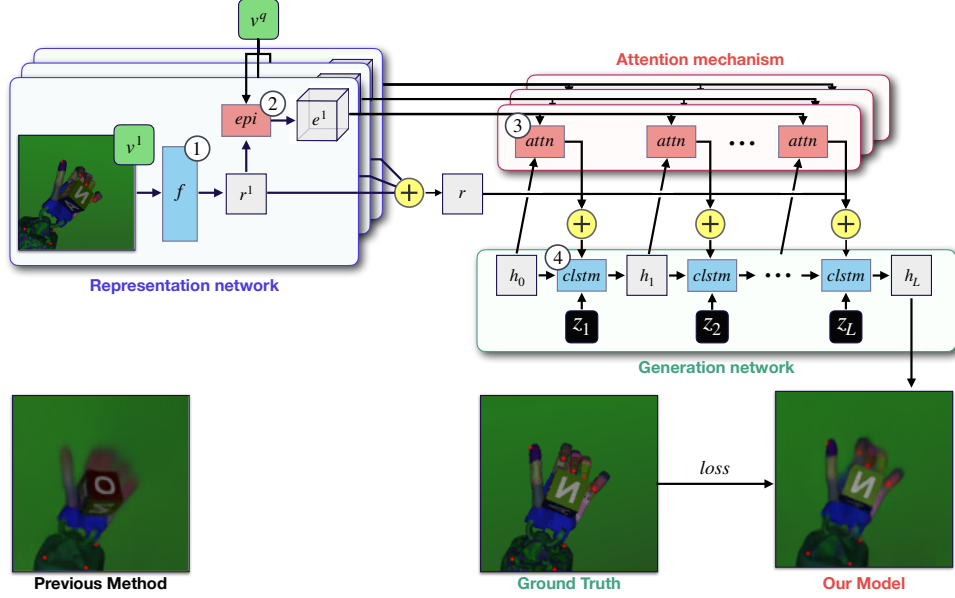

Figure 1: Overview of the model architecture used in our experiments. Grey boxes denote intermediate representations, $z_i$ latent variables, $+$ element-wise addition, and blue and red boxes subcomponents of the neural network architecture. Green components are model inputs, blue are as in GQN [7], and red are contributions of our model. (1) Context images and corresponding viewpoints are passed through a convolutional neural network $f$ to produce a context representation. We use the Tower architecture from [7]. (2) We use the epipolar geometry between the query viewpoint and the context viewpoint to extract the features in the context representation $r^k$ that are relevant to rendering each spatial point in the query viewpoint $v^q$. These extracted features are stored in a 3-dimensional tensor $e^k$ called the epipolar representation. See Figure 3(a) for more details. (3) At each generation step $l$, we compute the decoder input by attending over the epipolar representation. The attention map $a_l$ captures the weighted contribution to each spatial position in the decoder hidden state of all relevant positions in the context representations. See Figure 3(b) for more details on how the attention map is computed. (4) The decoder, or generation network, is the skip-connection convolutional LSTM cell from [7]. It takes as input the attention map $a^l$, previous hidden state $h_{l-1}$, and a latent variable $z_l$, which is used to model uncertainty in the predicted output. See [7] for more details.

Unlike GQN, GQN with ECA (E-GQN) can model relationships between pixels that are spatially distant in the context images, and can use a different representation at each layer in the decoder. And unlike more generic approaches to non-local attention, which require comparing each spatial location to every other spatial location ($O(n^2)$ comparisons per pixel for a $n \times n$ image), [42], E-GQN only requires each spatial location to be compared to a subset of the other spatial locations (requiring only $O(n)$ comparisons per pixel for a $n \times n$ image).

We evaluate our approach on datasets from the original GQN paper and three new datasets designed to test the ability to render systems with many degrees of freedom and a wide variety of objects. We find significant improvements in a lower bound on the negative log likelihood (the ELBO), per-pixel mean absolute error, and qualitative performance on most of these datasets.

To summarize, our key contributes are as follows:

1. We introduce a novel attention mechanism, Epipolar Cross-Attention (ECA), that leverages the geometry of the camera poses to perform efficient non-local attention.

2. We introduce three datasets: *Disco Humanoid*, *OpenAI Block*, and *Room-Random-Objects* as a testbed for neural rendering with complex objects and high-dimensional state.

3. We demonstrate the ECA with GQN (E-GQN) improves neural rendering performance on those datasets.

## 2 Background

### 2.1 Problem description

Given $K$ images $x^k \in X$ and corresponding camera viewpoints $v^k \in V$ of a scene $s$, the goal of neural rendering is to learn a model that can accurately predict the image $x^q$ the camera would see from a query viewpoint $v^q$. More formally, for distributions of scenes $p(S)$ and images with corresponding viewpoints $p(V, X \mid s)$, the goal of neural rendering is to learn a model that maximizes

$$\mathbb{E}_{s \sim p(S)} \mathbb{E}_{v^q, x^q \sim p(V, X \mid s)} \mathbb{E}_{v^k, x^k \sim p(V, X \mid s)} \log p\left(x^q | (x^k, v^k)_{k=\{1, \cdots, K\}}, v^q\right)$$

This can be viewed as an instance of few-shot density estimation [29].

### 2.2 Generative Query Networks

Generative Query Networks [7] model the likelihood above with an encoder-decoder neural network architecture. The encoder, or *representation network* is a convolutional neural network that takes $v^k$ and $x^k$ as input and produces a representation $r$.

The decoder, or *generation network*, takes $r$ and $v^q$ as input and predicts the image rendered from that viewpoint. Uncertainty in the output is modeled using stochastic latent variables $z$, producing a density $g(x^q \mid v^q, r) = \int g(x^q, z \mid v^q, r) dz$ that can be approximated tractably with a variational lower bound [20, 7]. The generation network architecture is based on the skip-connection convolutional LSTM decoder from DRAW [12].

### 2.3 Epipolar Geometry

The epipolar geometry between camera viewpoints $v^1$ and $v^2$ describes the geometric relationship between 3D points in the scene and their projections in images $x^1$ and $x^2$ rendered from pinhole cameras at $v^1$ and $v^2$ [14]. Figure 2 describes the relationship.

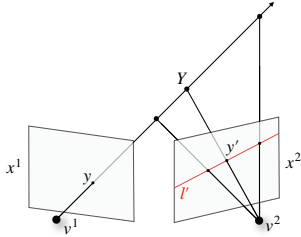

Figure 2: Illustration of the epipolar geometry. For any image point $y$ in $x^1$ corresponding to a 3D point $Y$, the image point $y'$ in $x^2$ that corresponds to $Y$ lies on a line $\mathbf{l}'$ in $x^2$. This line corresponds to the projection onto $x^2$ of the ray passing through the camera center of $v^1$ and $y$ and depends only on the intrinsic geometry between $v^1$ and $v^2$, not the content of the scene.

There is a linear mapping called the *fundamental matrix* that captures this correspondence [14]. The fundamental matrix is a mapping $F$ from an image point $y$ in $x^1$ to the epipolar line $\mathbf{l}'$. $F$ is a $3 \times 3$ matrix that depends on $v^1$ and $v^2$. The image point $y' = (h', w', 1)$ lies on the line $\mathbf{l}'$ corresponding to $y$ if $y'$ lies on the line $ah' + bw' + c = 0$, where $Fy = [a, b, c]^T$.

## 3 Epipolar Cross-Attention

In GQN, the scene representation is an element-wise sum of context representations from each context viewpoint. The context representations are created from the raw camera images through convolutional layers. Since convolutions are local operations, long-range dependencies are difficult to model [15, 16, 42]. As a result, information from distant image points in the context representation may not propagate to the hidden state of the generation network.

The core idea of Epipolar Cross-Attention is to allow the features at a given spatial position $y$ in the generation network hidden state to depend directly on all of the relevant spatial positions in the context representations. Relevant spatial positions are those that lie on the epipolar line corresponding to $y$ in each context viewpoint.

Figure 1 describes our model architecture. Our model is a variant of GQN [7]. Instead of using $r = \sum_k r^k$ as input to compute the next generation network hidden state $h_l$, we use an attention

map computed using our epipolar cross-attention mechanism. The next two subsections describe the attention mechanism.

## 3.1 Computing the Epipolar Representation

For a given spatial position $y = (p_0, p_1)$ in the decoder hidden state $h_l$, the epipolar representation $e^k$ stores at $(p_0, p_1)$ all of the features from $r^k$ that are relevant to rendering the image at that position.[2]

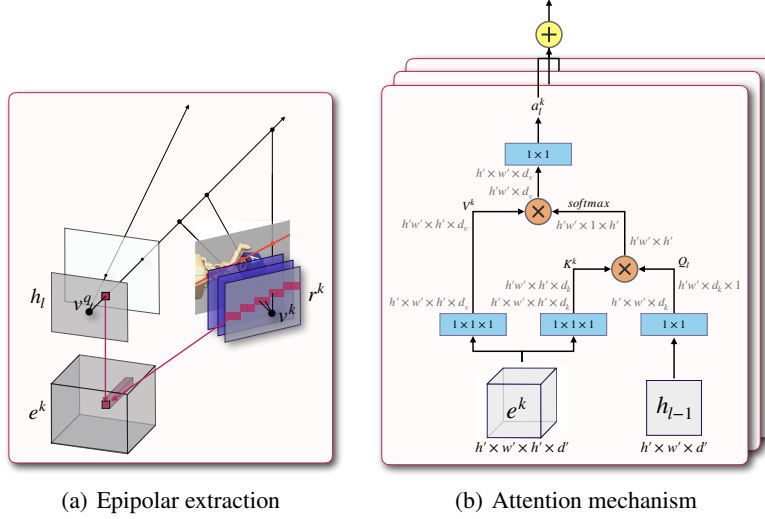

(a) Epipolar extraction       (b) Attention mechanism

Figure 3: (a) Constructing the epipolar representation $e^k$ for a given camera viewpoint $v^k$. For a given spatial position in the decoder state $h_l$, there is a 1-dimensional subset of feature maps $l'$ in $r^k$ arising from the epipolar geometry. This can be viewed as the projection of the line passing from the camera center at $v^q$ through the image point onto $r^k$. The epipolar representation $e^k$ is constructed by stacking these lines along a third spatial dimension. Note that $h_l$ and $r_k$ are $h' \times w' \times d'$ feature maps, so $e^k$ has shape $h' \times w' \times h' \times d'$. If $h = w$ is the size of the image, $h' = w' = h/4$ and $d' = 256$ in our experiments. (b) Our attention mechanism. Blue rectangles denote convolutional layers with given kernel size. "×" denotes batch-wise matrix multiplication, and "+" element-wise summation. The previous decoder hidden state $h_{l-1}$ is used to compute a query tensor $Q_l$ by linear projection. The epipolar representation $e^k$ is also linearly projected to compute a key tensor $K^k$ and value tensor $V^k$. $K^k$ and $Q_l$ are matrix-multiplied to form unnormalized attention weights, which are scaled by $1/\sqrt{d_k}$. A softmax is computed along the final dimension, and the result is multiplied by $V^k$ to get an attention score as in [41]. All of the attention scores are linearly projected into the correct output dimension and summed element-wise.

Figure 3(a) shows how we construct the epipolar representation. To compute the epipolar line $\mathbf{l}'_y$ in $r^k$, we first compute the fundamental matrix $F_q^k$ arising from camera viewpoints $v^q$ and $v^k$ [14], and then find $\mathbf{l}'_y = F_q^k[p_0, p_1, 1]^T$.

If $h_l$ has shape $(h', w')$, then for each $0 \leq p'_1 < w'$,

$$e^k_{p_0, p_1, p'_1} = r^k_{p'_0, p'_1}$$

where the subscripts denote array indexing and $p'_0$ is the point on $\mathbf{l}'_y$ corresponding to $p'_1$.[3]

All of these operations can be performed efficiently and differentiably in automatic differentiation libraries like Tensorflow [1] as they can be formulated as matrix multiplication or gather operations.

## 3.2 Attention mechanism

Figure 3(b) describes our attention mechanism in more detail. We map the previous decoder hidden state $h_{l-1}$ and the epipolar representations $e^k$ to an attention score $a_l^k$. $a_l^k$ represents the weighted contribution to each spatial position of all of the geometrically relevant features in the context representation $r^k$.

Typically the weights for the projections are shared between context images and decoder steps. To facilitate passing gradients to the generation network, the attention maps $a_l^k$ are provided a skip connection to $r^k$, producing

$$a_l = \lambda \sum_k a_l^k + \sum_k r^k$$

.

where $\lambda$ is a learnable parameter. $a_l$ is used as input to to produce the next hidden state $h_l$.

# 4 Experiments

## 4.1 Datasets

To evaluate our proposed attention mechanism, we trained GQN with Epipolar Cross-Attention (E-GQN) on four datasets from the GQN paper: Rooms-Ring-Camera (RRC), Rooms-Free-Camera (RFC), Jaco, and Shepard-Metzler-7-Parts (SM7) [7, 35]. We chose these datasets for their diversity and suitability for our method. Other datasets are either easier versions of those we used (Rooms-Free-Camera-No-Object-Rotations and Shepard-Metzler-5-Parts) or focus on modeling the room layout of a large scene (Mazes). Our technique was designed to help improve detail resolution in scenes with high degrees of freedom and complex objects, so we would not expect it to improve performance in an expansive, but relatively low-detail dataset like Mazes.

The GQN datasets are missing several important features for robotic representation learning. First, they contain only simple geometric objects. Second, they have relatively few degrees of freedom: objects are chosen from a fixed set and placed with two positional and 1 rotational degrees of freedom. Third, they do not require generalizing to a wide range of objects. Finally, with the exception of the Rooms-Free-Camera dataset, all images are size $64 \times 64$ or smaller.

To address these limitations, we created three new datasets: OpenAI Block (OAB), Disco Humanoid (Disco), and Rooms-Random-Objects (RRO) [4]. All of our datasets are rendered at size $128 \times 128$. Examples from these datasets are shown alongside our model's renderings in Figure 6.

The OAB dataset is a modified version of the domain randomized [38] in-hand block manipulation dataset from [28, 27] where cameras poses are additionally randomized. Since this dataset is used for sim-to-real transfer for real-world robotic tasks, it captures much of the complexity needed to use neural rendering in real-world robotics, including a 24-DoF robotic actuator and a block with letters that must be rendered in the correct 6-DoF pose.

The Disco dataset is designed to test the model's ability to accurately capture many degrees of freedom. It consists of the 27-DoF MuJoCo [39] model from OpenAI Gym [3] rendered with each of its joints in a random position in $[-\pi, \pi)$. Each of the geometric shape components of the Humanoid's body are rendered with a random simple texture.

The RRO dataset captures the ability of models to render a broad range of complex objects. Scenes are created by sampling 1-3 objects randomly from the ShapeNet [4] object database. The floor and walls of the room as well as each of the objects are rendered using random simple textures.

## 4.2 Experimental setup

We use the the "Tower" representation network from [7]. Our generation network is from Figure S2 of [7] with the exception of our attention mechanism. The convolutional LSTM hidden state and skip connection state have 192 channels. The generation network has 12 layers and weights are

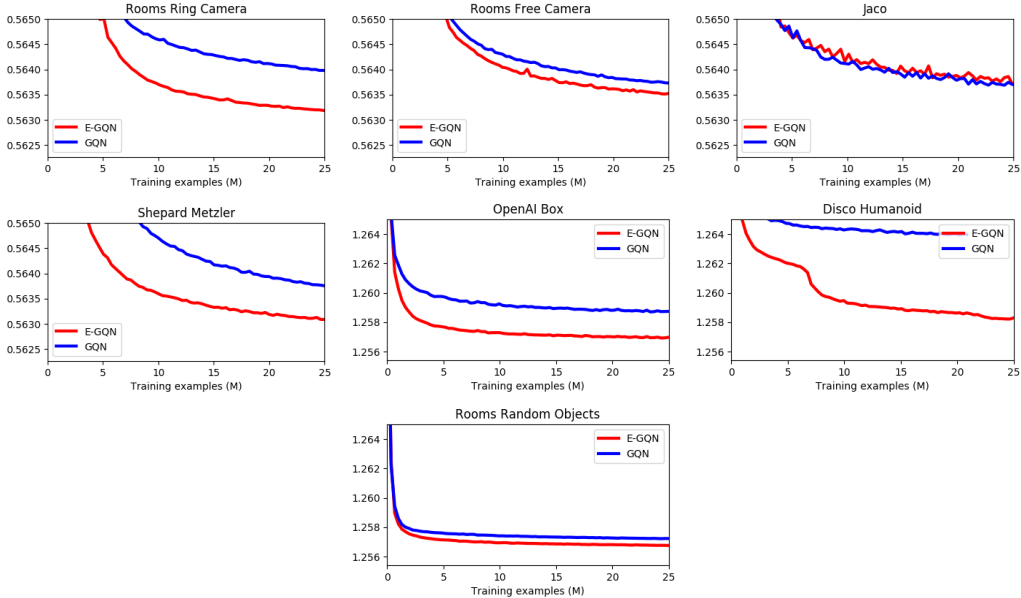

Figure 4: ELBO (nats/dim) on the test set. The minimum y-value denotes the theoretical minimum error. We compute this value by setting the KL term to 0 and the mean of the output distribution to the true target image. Note that this value differs for the GQN datasets and ours because we use a different output variance on our datasets as discussed in Section 4.2.

shared between generation steps. We always use 3 context images. Key dimension $d_k = 64$ for all experiments, and value dimension $d_v = 128$ on the GQN datasets with $d_v = 192$ on our datasets.

We train our models using the Adam optimizer [19]. We ran a small hyperparameter sweep to choose the learning rate schedule and found that a learning rate of 1e-4 or 2e-4 linearly ramped up from 2e-5 over 25,000 optimizer steps and then linearly decayed by a factor of 10 over 1.6M optimizer steps performs best in our experiments.

We use a batch size of 36 in experiments on the GQN datasets and 32 on our datasets. We train our models on 25M examples on 4 Tesla V-100s (GQN datasets) or 8 Tesla V-100s (our datasets).

As in [7], we evaluate samples from the model with random latent variables, but taking the mean of the output distribution. Input and output images are scaled to $[-0.5, 0.5]$ on the GQN datasets and $[-1, 1]$ on ours. Output variance is scaled as in [7] on the GQN datasets but fixed at 1.4 on ours.

## 4.3 Quantitative results

| Dataset | Mean Absolute Error (pixels) | | Root Mean Squared Error (pixels) | | ELBO (nats / dim) | |
|---|---|---|---|---|---|---|
| | GQN | E-GQN | GQN | E-GQN | GQN | E-GQN |
| rrc | $7.40 \pm 6.22$ | $\mathbf{3.59 \pm 2.10}$ | $14.62 \pm 12.77$ | $\mathbf{6.80 \pm 5.23}$ | $0.5637 \pm 0.0013$ | $\mathbf{0.5629 \pm 0.0008}$ |
| rfc | $12.44 \pm 12.89$ | $\mathbf{12.05 \pm 12.79}$ | $\mathbf{26.80 \pm 21.35}$ | $27.65 \pm 20.72$ | $\mathbf{0.5637 \pm 0.0011}$ | $0.5639 \pm 0.0012$ |
| jaco | $4.30 \pm 1.12$ | $\mathbf{4.00 \pm 0.90}$ | $8.58 \pm 2.94$ | $\mathbf{7.43 \pm 2.32}$ | $0.5634 \pm 0.0007$ | $\mathbf{0.5631 \pm 0.0005}$ |
| sm7 | $3.13 \pm 1.30$ | $\mathbf{2.14 \pm 0.53}$ | $9.97 \pm 4.34$ | $\mathbf{5.63 \pm 2.21}$ | $0.5637 \pm 0.0009$ | $\mathbf{0.5628 \pm 0.0004}$ |
| oab | $10.99 \pm 5.13$ | $\mathbf{5.47 \pm 2.54}$ | $22.11 \pm 8.00$ | $\mathbf{10.39 \pm 4.55}$ | $1.2587 \pm 0.0018$ | $\mathbf{1.2569 \pm 0.0011}$ |
| disco | $18.86 \pm 7.16$ | $\mathbf{12.46 \pm 9.27}$ | $32.72 \pm 6.32$ | $\mathbf{22.04 \pm 11.08}$ | $1.2635 \pm 0.0055$ | $\mathbf{1.2574 \pm 0.0007}$ |
| rro | $10.12 \pm 5.15$ | $\mathbf{6.59 \pm 3.23}$ | $19.63 \pm 9.14$ | $\mathbf{12.08 \pm 6.52}$ | $1.2573 \pm 0.0011$ | $\mathbf{1.2566 \pm 0.0009}$ |

Figure 5: Performance of GQN and E-GQN. Note: ELBO scaling is due to different choices of output variance as discussed in Figure 4.

Figure 4 shows the learning performance of our method. Figure 5 shows the quantitative performance of the model after training. Both show that our method significantly outperforms the baseline on most datasets, with the exception of Jaco and RFC, where both methods perform about the same.

### 4.4 Qualitative results

Figures 6 shows randomly chosen samples rendered by our model on our datasets. On OAB, our model near-perfectly captures the pose of the block and hand and faithfully reproduces their textures, whereas the baseline model often misrepresents the pose and textures. On Disco, ours more accurately renders the limbs and shadow of the humanoid. On RRO, ours faithfully (though not always accurately) renders the shape of objects, whereas the baseline often renders the wrong object in the wrong location. Quality differences are more subtle on the original GQN datasets.

For more examples, including on those datasets, see the website for this paper [5].

### 4.5 Discussion

E-GQN improves quantitative and qualitative neural rendering performance on most of the datasets in our evaluations. We hypothesize that the improved performance is due to the ability of our model to query features from spatial locations in the context images that correspond in 3D space, even when those spatial locations are distant in pixel space.

Our model does not improve over the baseline for the Jaco and RFC datasets. Jaco has relatively few degrees of freedom, and both methods perform well. In RFC, since the camera moves freely, objects contained in the target viewpoint are usually not contained in context images. Hence the lack of performance improvement on RFC is consistent with our intuition that E-GQN helps when there are 3D points contained in both the context and target viewpoints.

There are two performance disadvantages of our implementation of E-GQN. First, E-GQN requires computing the epipolar representation $e^k$ for each context viewpoint. Each $e^k$ is a $h' \times w' \times h' \times d'$ tensor, which can could cause issues fitting $e^k$ into GPU memory for larger image sizes. Second, due to extra computation of $e^k$ and the attention maps $a_l$, E-GQN processes around 30% fewer samples per second than GQN in our experiments. In practice, E-GQN reaches a given loss value significantly faster in wall clock time on most dastasets due to better data efficiency.

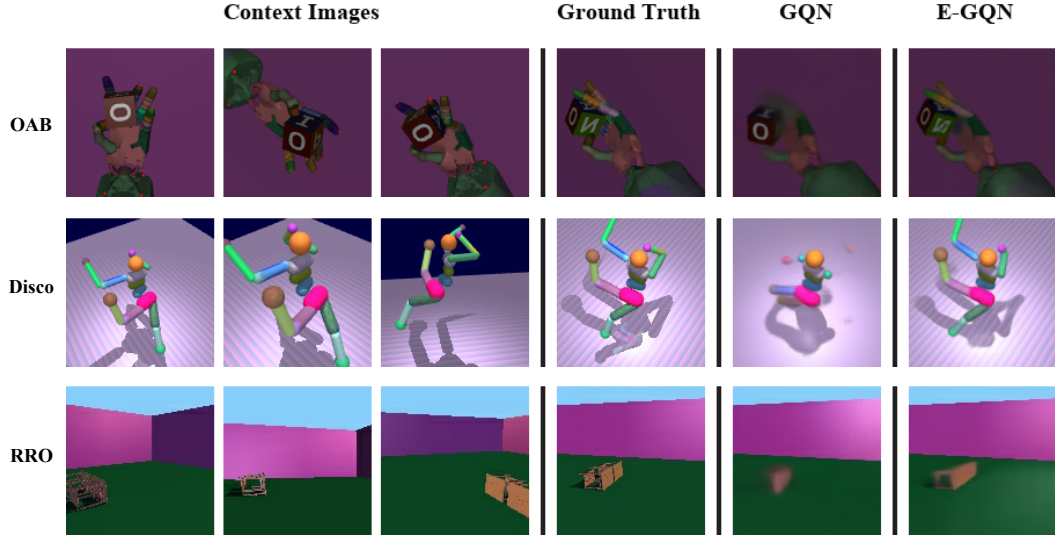

Figure 6: Images rendered by our model. See the website for this paper[6] for more examples.

## 5 Related work

### 5.1 Multi-view 3D reconstruction

Constructing models of 3D scenes from multiple camera views is a widely explored subfield of computer vision. If the camera poses are unknown, Structure-from-Motion (SfM) techniques [32, 2]

(for unordered images) or Simultaneous Localization and Mapping (SLAM) techniques [5] (for ordered images from a real-time system) are typically used. If camera poses are known, multi-view stereo or multi-view reconstruction (MVR) can be applied.

MVR techniques differ in how they represent the scene. Voxels [6], level-sets [8], depth maps [36], and combinations thereof are common [33]. They also differ in how they construct the scene representation. Popular approaches include adding parts that meet a cost threshold [34], iteratively removing parts that do not [22, 10], or fitting a surface to extracted feature points [26].

Most MVR techniques do not rely on ground truth scene representations and instead depend on some notion of consistency between the generated scene representation and the input images like scene space or image space photo consistency measures [18, 22, 33].

### 5.2 Deep learning for 3D reconstruction

Recently, researchers have used deep learning to learn the mapping from images to a scene representation consisting of voxels [40, 44, 43, 30, 45] or meshes [30], with supervisory signal coming from verifying the 3D volume against known depth images [40, 45] or coming from a large-scale 3D model database [44, 30]. Loss functions include supervised losses [45], generative modeling objectives [30], a 3D analog of deep belief networks [23, 44], and a generative adversarial loss [43, 11].

Some neural network approaches to 3D understanding instead create implicit 3D models of the world. By training an agent end-to-end using deep reinforcement learning [25] or path planning and imitation learning [13], agents can learn good enough models of their environments to perform tasks in them successfully. Like our work, Gupta and coauthors also incorporate geometric primitives into their model architecture, transforming viewpoint representations into world coordinates using spatial transformer layers [17]. Instead of attempting to learn 3D representations that help solve a downstream task, other approaches learn generic 3D representations by performing multi-task learning on a variety of supervised learning tasks like pose estimation [46].

### 5.3 View Synthesis and neural rendering

Neural rendering or view synthesis approaches learn an implicit representation of the 3D structure of the scene by training a neural network end-to-end to render the scene from an unknown viewpoint. In [37], the authors map an images of a scene to an RGB-D image from an unknown viewpoint with an encoder-decoder architecture, and train their model using supervised learning. Others have proposed incorporating the geometry of the scene into the neural rendering task. In [9], plane-sweep volumes are used to estimate depth of points in the scene, which are colored by a separate network to perform view interpolation (i.e., the input and output images are close together). Instead of synthesizing pixels from scratch, other work explores using CNNs to predict appearance flow [47].

In [7], the authors propose the generative query network model (GQN) model architecture for neural rendering. Previous extensions to GQN include augmenting it with a patch-attention mechanism [31] and extending it to temporal data [21].

## 6 Conclusion

In this work, we present a geometrically motivated attention mechanism that allows neural rendering models to learn more accurate 3D representations and scale to more complex datasets with higher dimensional images. We show that our model outperforms an already strong baseline. Future work could explore extending our approach to real-world data and higher-dimensional images.

The core insight of this paper is that injecting geometric structure ino neural networks can improve conditional generative modeling performance. Another interesting direction could be to apply this insight to other types of data. For example, future work could explore uncalibrated or moving camera systems, video modeling, depth prediction from stereo cameras, or constructing explicit 3D models.

## Footnotes

[2]Note that care must be taken that the representation network does not change the effective field of view of the camera.

[3]To make sure $p'_0$ are valid array indices, we round down to the nearest integer. For indices that are too large or too small, we instead use features of all zeros.

[4]Our datasets are available here: https://github.com/josh-tobin/egqn-datasets

[5]https://sites.google.com/view/geometryaware-neuralrendering/home

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
