[Supplementary Material]

# A   Example images rendered using our method

This section contains additional example images rendered using GQN and our method.

Figure 7: Rooms Ring Camera

Figure 8: Rooms Free Camera

Figure 9: Jaco

Figure 10: Shepard-Metzler 7 Parts

Figure 11: OpenAI Box.

**Context Images**      **Ground Truth**    **GQN**      **E-GQN**

Figure 12: Disco Humanoid

**Context Images**      **Ground Truth**    **GQN**      **E-GQN**

Figure 13: Rooms Random Objects