[Reviews · NeurIPS 2019]

Reviewer 1



Originality I think this is very original, and is an elegant way of adding the known physical process of image formation to the deep learning setup that GQN proposes. This is a natural yet original extension. Quality This submission is high quality. The problem is well presented and explained, the link to epipolar geometry feels very smooth and the way that it is incorporated in the system is clean. This works is self-contained, and well presented. Clarity The paper is well written and all the concepts are very clear, I have nothing negative to report on clarity. Significance This is quite significant as it proposes a tool that incorporates epipolar geometry to a deep learning architecture, and it applicable beyond the scope of GQN. If the camera intrinsics and extrinsics are known in a multi-view problem such as the one presented in this paper, I do not see a reason why this technique should not be used.

Reviewer 2



Epipolar geometry describes the relationship between views of the same scene captured by a pinhole camera. The authors refer to the classic reference Hartley & Zisserman's Multiview Geometry book for the details, but clearly explain the fundamental properties they leverage in the proposed scene encoder design. Namely, given two calibrated views of the same scene, for each point in one view, if a match can be found in the other view, it will be located along a line whose equation is a relatively simple function of the corresponding relative camera poses. That is, if the relative pose of two cameras is known, the search for a matching point between two frames is a 1D search problem. The authors exploit this fact to reduce the amount of data that a GQN scene encoder needs to ignore/discard when merging information from multiple view into a sensible scene representation, in two steps. After the context frames are encoded, given the query camera: * epipolar lines are extracted and stacked; * search is implemented as an attention mechanism. By using this architecture the encoder doesn't have to learn to approximate multi-view geometry and can use its capacity to solve a simpler matching problem and to fuse information into a more coherent representation. The architecture is described in full detail, and even without source code I am confident the new model could be reimplemented fairly easily (in comparison reimplementing Conv-DRAW would be much harder). The authors show that EGQN can consistently outperform vanilla GQN on a subset of the original benchmarks, and that with the new architecture let them scale the model to capture more complex scene geometries. To this end, the authors introduce 3 novel datasets, and commit to make them available (although the website the point to for extra results in not operative). One aspect of the experimental setup is disappointing: the Maze dataset (available alongside the other datasets) is not used for comparisons. I would have expected this setup to be particularly interesting, and a more difficult one for EGQN as many of the frames are collected by RL agents walking down corridors. The peculiar relative pose of forward motion is a corner case that perhaps cannot be handled by the model (specific implementation)? There is one detail of the presentation that I found confusing: the abstract at line 7 reads 'requiring only O(n) comparisons per spatial dimension instead of O(n^2)'. The statement is reiterated and somewhat clarified at line 34, but then it's not further discussed in the text. Whilst I do have an intuition of what the author means, I think it would be helpful to explicitly clarify the subject in the discussion of the algorithm.

Reviewer 3



The idea is original, the paper aims to improve a simple element-wise sum of context representations as in GQN. The key idea is based that there should be a consistency between the viewpoints, i.e. for a given pixel we can cast a ray from the pinhole camera, then from a different viewpoint this pixel colour should lie somewhere on the casted ray (yet can be occluded). The output of the model should then depend only on the relevant features, i.e. for a given pixel only on the rays that pass though it. The overall network and attention mechanism (Fig2 and Fig3 b) is complex and have some extra unexplained engineering. I did not understand the approach properly from the descriptions. For example is e^k aggregated for all observations, or separate, is the attention mechanism looking at all of them together? Where z_i variables come from, seems these are not specified. In Fig2 it is unclear which operations are done per observation and which across all. In Fig3b, what is d, what is its value any why? Why e^k dim is 2 times h' and one time w' if h/w (height/width) of an image should be symmetric? Regarding the results, the EGQN results on the new datasets are much better than GQN. This is especially visible in Fig6 where only EGQN gets the colours and letters of the box correctly. Similarly, EGQN performs well on disco humanoid, and GQN performs poorly. The original datasets are very simple and here the results are similar, only sm7 is more challenging from the original ones are here the performance is again much better. Fig8 ground truth is the same as the last column of the context, and predictions are different, it seems GT is incorrect. Would be good to explain each variable in the caption of each figure, not only in the main text. The evaluation is only one table with 2 rows. Would be good to show other measures, and for this table, the presentation could be richer, e.g. show violin plots with std/median depicted too. Surprisingly exemplary images are not given for all datasets at least in the supplement. Fig4: What is +1.254 on top of 3 plots meaning? Fig4 minimum y-value looks suspicious -- why is the value constant for 4 datasets and then for the 3 other ones? The method is interesting overall but evaluation seems to be not adequately explored, visualised, and presented or even provided. There are some issues raised that hopefully will be explained in the rebuttal.

[Author Response · NeurIPS 2019]

We thank the reviewers for their helpful feedback and positive view of the work. We will aim to incorporate your
suggestions in future versions of the paper.
**Re: dataset release.** We are planning to release the new datasets in the next few weeks.

**Reviewer 1**

• **Add a speed comparison**. E-GQN usually reaches a given loss significantly (often 10x) faster in wall clock time (in
addition to having a lower final loss). On the two datasets (rfc, jaco) where final performance is similar, E-GQN is
somewhat slower due to processing $\sim 30\%$ fewer samples per second. We will add a speed analysis to the final paper.
• **Provide visualization of the attention mechanism.** If accepted, we will add one to the supplemental materials.
• **Add an ablation study**. Since the aim of our work is to design & study a new attention mechanism, we hold the
remainder of the architecture constant, so the GQN numbers show the performance without the attention mechanism.

**Reviewer 2**

• **Discuss how approaches similar in spirit to EGQN could benefit training of conditional models in general.**
Good question. Geometric structure could also be exploited in settings with moving cameras. Temporal locality may
also be exploitable. We will expand on these in the conclusion.
• **Have the authors analyzed EGQN's scene representation?** We agree this would be interesting, but considered it
out of scope for this paper as our main aim was improving model performance on more challenging datasets.
• **Discuss weaknesses or in general cons of the proposed model.** Thanks for pointing this out. The main weaknesses
of the method are inapplicability when there is little content overlap between context and target frames (which we
briefly mention in the paper), additional memory requirement relative to GQN, and slower training in terms of samples
per second (though usually faster training overall as mentioned above). We will expand in the paper.

**Reviewer 3**

• **Would be good to show other evaluation measures.** We updated the evaluation to include standard deviation,
RMSE, and ELBO in addition to existing evalution for all 7 datasets. The new table is included below as **Figure 1**.
• **Exemplary images are not given for all datasets.** Thanks for pointing this out. We agree more images would help
readers better understand the models' performance. We uploaded additional images from all datasets to the supplemental
website for the paper, and also included a few of them below as **Figure 2**.
• **How was the min Y value in Fig 4 computed?** We compute the minimum ELBO by assuming the KL term to be 0
and the mean of the output distribution to be the true target image. Per Section 4.2, we use a different output variance
hyperparameter for our datasets vs. those from the original GQN paper, which causes the different scaling of the ELBO.
We will add more detail to the description in the paper.
• **Would be nice to compare to other methods that build on the GQN, e.g. CGQN.** CGQN addresses the problem
of multiple simultaneous predictions being inconsistent with one another. Since we focused on improving the quality of
individual predictions, we considered it out of scope. Combining CGQN and E-GQN could be interesting future work.

| Dataset | Mean Absolute Error (pixels) | | Root Mean Squared Error (pixels) | | ELBO (nats / dim) | |
|---|---|---|---|---|---|---|
| | GQN | E-GQN | GQN | E-GQN | GQN | E-GQN |
| rrc | $7.40 \pm 6.22$ | $\mathbf{3.59 \pm 2.10}$ | $14.62 \pm 12.77$ | $\mathbf{6.80 \pm 5.23}$ | $0.5637 \pm 0.0013$ | $\mathbf{0.5629 \pm 0.0008}$ |
| rfc | $12.44 \pm 12.89$ | $\mathbf{12.05 \pm 12.79}$ | $\mathbf{26.80 \pm 21.35}$ | $27.65 \pm 20.72$ | $\mathbf{0.5637 \pm 0.0011}$ | $0.5639 \pm 0.0012$ |
| jaco | $4.30 \pm 1.12$ | $\mathbf{4.00 \pm 0.90}$ | $8.58 \pm 2.94$ | $\mathbf{7.43 \pm 2.32}$ | $0.5634 \pm 0.0007$ | $\mathbf{0.5631 \pm 0.0005}$ |
| sm7 | $3.13 \pm 1.30$ | $\mathbf{2.14 \pm 0.53}$ | $9.97 \pm 4.34$ | $\mathbf{5.63 \pm 2.21}$ | $0.5637 \pm 0.0009$ | $\mathbf{0.5628 \pm 0.0004}$ |
| oab | $10.99 \pm 5.13$ | $\mathbf{5.47 \pm 2.54}$ | $22.11 \pm 8.00$ | $\mathbf{10.39 \pm 4.55}$ | $1.2587 \pm 0.0018$ | $\mathbf{1.2569 \pm 0.0011}$ |
| disco | $18.86 \pm 7.16$ | $\mathbf{12.46 \pm 9.27}$ | $32.72 \pm 6.32$ | $\mathbf{22.04 \pm 11.08}$ | $1.2635 \pm 0.0055$ | $\mathbf{1.2574 \pm 0.0007}$ |
| rro | $10.12 \pm 5.15$ | $\mathbf{6.59 \pm 3.23}$ | $19.63 \pm 9.14$ | $\mathbf{12.08 \pm 6.52}$ | $1.2573 \pm 0.0011$ | $\mathbf{1.2566 \pm 0.0009}$ |

Figure 1: Performance of GQN and E-GQN. Note: ELBO scaling is due to different choices of output variance.

Figure 2: Randomly chosen samples from GQN and E-GQN

[Meta-Review · NeurIPS 2019]

All reviewers agree that the paper is of high quality and will be of interest to the NeurIPS community. I strongly encourage the authors to include results on the Maze dataset as it has been brought up by two of the reviewers. I look forward to reading the camera ready version of the manuscript.